# CellTarNet: Single-Cell Perturbation Prediction using Transformer based Normalizing Flow

**Shiv Shankar**
University of Massachusetts
USA
sshankar@cs.umass.edu

## Abstract

Predicting the transcriptional response of cells to perturbations is a challenging task as perturbation datasets often include global gene expression shifts, which are difficult to seperate from individual expression. We present CellTarNet, a generative framework based on transformer based normalizing flows to learn transport from control cells to perturbed cells. The transformer encoder summarizes control-cell states into a context representation, while the normalizing flow learns a distribution over perturbed transcriptional profiles conditioned on the context. We employ contrastive matching to pull predicted samples toward the true perturbed distribution and separates them from mismatched perturbation-context pairs. We further show how to integrate this with gene interaction graphs to better model gene expressions. CellTarNet outperforms prior methods across multiple benchmarks.

## 1 Introduction

The ability to accurately predict a cell's transcriptomic response to a genetic knockout or a chemical compound is an important challenge in computational biology (Lotfollahi et al., 2023). In-silico models (i.e. using computer models) for estimating perturbation outcomes carry considerable promise for accelerating the design of novel therapeutics by enabling the rapid virtual screening of drug candidates against specific disease cell states (Qi et al., 2024; Viñas Torné et al., 2025). Furthermore these are also important for individualized medicine where the treatments could be adapted to an individual's unique profile (Wang et al., 2025). However, the traditional, purely experimental approach to this problem of physically perturbing cells and measuring the outcome through high-throughput assays is expensive, time-consuming, and fundamentally non-scalable at the level of individual patients. Consequently, the development of models capable of predicting the outcomes of unseen perturbations is critical for the next generation of biomedical discovery.

The biggest challenge in building such models stems from the basics of single-cell sequencing data. Current methods render it physically impossible to observe the same biological cell both before and after a perturbation. Instead, we must work with measurements from distinct, but matched, populations of control and perturbed cells. This reality means the task is more complex than a vanilla regression problem: the predictions should focus not only on mean shifts but also the broader response of cells to a specific intervention. This is important because cellular responses are rarely uniform (Song et al., 2025). Perturbations can induce complex, higher-order effects, such as bifurcating cell fate pathways, increasing population heterogeneity, or selective effects on a vulnerable subgroup that may be therapeutically relevant (Mejia et al., 2025; Yu et al., 2025; Theodoris et al., 2023).

Recent advances have seen the application of large-scale foundation models, such as Geneformer (Cui et al., 2024b) and scGPT (Cui et al., 2024a), which are pre-trained on massive, diverse single-cell atlases. While these models have shown promise for biological understanding, empirical analyses by Viñas Torné et al. (2025); Ahlmann-Eltze et al. (2025); Kedzierska et al. (2025) demonstrate that these models often underperform simpler, well-calibrated baselines on the task of perturba-

tion prediction, particularly when standard batch correction techniques are applied to the simpler models. This suggests that general pre-training may not be sufficient for this precise inductive task (Kedzierska et al., 2025). Additionally, models that treat the genes as independent inputs often fail to generalize to unseen cell types or out-of-distribution perturbations, as they do not capture the context-dependent gene-gene interaction networks that govern transcriptional regulation (Aguirre et al., 2025; Young et al., 2016).

To address these challenges, we propose a novel method that combines two powerful ideas. First, we adopt a TARNet (Zhai et al., 2024) style architecture at its core, which is naturally suited for modeling heterogeneous multimodal distributions. This is coupled with a contrastive training (Oord et al., 2018) objective that explicitly operates at the population level (Lee et al., 2024; Elsharkawy & Kahn, 2025). Instead of forcing noisy one-to-one correspondences between individual cells, this formulation encourages the model to align the entire latent distribution of the generated perturbed cells with that of the true perturbed population (Parulekar et al., 2023). This allows the model to capture not just the mean shift but the full shape of the response distribution, including changes in variance and modality. Second, the transformer based backbone natively integrates prior biological knowledge. By incorporating knowledge about dependencies, such as gene regulatory networks, protein-protein interaction or co-expression networks, as an additional bias, we modulate the model's self-attention mechanism. This design prior guides the model to emphasize biologically plausible gene-gene relationships, enabling it to learn context-specific regulatory logic (Young et al., 2016; Aguirre et al., 2025).

## 2 PRELIMINARIES

### 2.1 NORMALIZING FLOWS

Normalizing Flows (NFs) (Rezende & Mohamed, 2015) represent a class of probabilistic generative models designed to construct complex probability distributions through a series of invertible, differentiable transformations applied to a simple base distribution. The core objective is to perform exact density estimation and sample generation, enabling tasks like data generation, variational inference, and likelihood-based training.

Mathematically, let $\mathbf{Z} \in \mathbb{R}^d$ be a random variable with a known, tractable probability density function (PDF) $p_Z(\mathbf{z})$ (e.g., a standard multivariate normal distribution). The goal is to model a target random variable $\mathbf{X} \in \mathbb{R}^d$ with an unknown complex distribution $p_X^*(\mathbf{x})$. An NF defines $\mathbf{X}$ as a transformation $T$ of $\mathbf{Z}$: $\mathbf{x} = T(\mathbf{z})$. If $T$ is bijective and both $T$ and its inverse $T^{-1}$ are differentiable (a *diffeomorphism*), the change-of-variables formula gives the exact PDF of $\mathbf{X}$:

$$p_X(\mathbf{x}) = p_Z(\mathbf{z}) \left| \det J_{T^{-1}}(\mathbf{x}) \right| = p_Z(T^{-1}(\mathbf{x})) \left| \det J_{T^{-1}}(\mathbf{x}) \right| \tag{1}$$

where $J_{T^{-1}}(\mathbf{x}) = \frac{\partial T^{-1}}{\partial \mathbf{x}}$ is the Jacobian matrix of the inverse transformation. The log-likelihood used for training is:

$$\log p_X(\mathbf{x}) = \log p_Z(T^{-1}(\mathbf{x})) + \log \left| \det J_{T^{-1}}(\mathbf{x}) \right| \tag{2}$$

The main challenge in NF design is creating a suitable $T$ that is simultaneously a) expressive enough to approximate complex distributions and b) has an easy to compute Jacobian determinant (Rezende & Mohamed, 2015).

For a general function $\mathbb{R}^d \to \mathbb{R}^d$, the Jacobian would require $O(d^3)$ time, which makes it unscalable for high-dimensional datasets. To address this $T$ is typically implemented as a composition of $K$ simpler, invertible layers : $T = T_K \circ T_{K-1} \circ \ldots \circ T_1$. where each layer has a simpler-to-compute Jacobian determinant (typically $O(d)$). This second requirement is usually addressed by using layers with triangular or diagonal Jacobians.

MASKED AUTOREGRESSIVE FLOW

Masked Autoregressive Flow (Papamakarios et al., 2017) or MAF is a powerful architecture design that combines the modeling capability of autoregressive models with the invertible transformation

framework of normalizing flows. It directly exploits the autoregressive structure inherent in sequential models to ensure that the layer has a triangular Jacobian, yielding a tractable determinant.

An autoregressive model factorizes the joint density of $\mathbf{x}$ as a product of conditionals:

$$p(\mathbf{x}) = \prod_{i=1}^{d} p(x_i|\mathbf{x}_{1:i-1}). \tag{3}$$

Each conditional is often modeled as a univariate distribution (e.g., Gaussian), whose parameters (mean $\mu_i$ and scale $\sigma_i$) are functions of the previous variables $\mathbf{x}_{1:i-1}$.

In MAF, the transformation from the base noise $\mathbf{z}$ to $\mathbf{x}$ is defined autoregressively:

$$z_i = \frac{x_i - \mu_i(\mathbf{x}_{1:i-1})}{\sigma_i(\mathbf{x}_{1:i-1})}. \tag{4}$$

This is the *inverse* pass (from $\mathbf{x}$ to $\mathbf{z}$). The scaling and shifting parameters ($\sigma_i$, $\mu_i$) are predicted by a neural network (an *autoregressive conditioner*) that takes $\mathbf{x}_{1:i-1}$ as input. A critical point to note is that for many neural architectures used in such sequential models these predictions for all $i$ can be made in a single forward pass by using masking as in the Masked Autoencoder for Distribution Estimation (MADE) network of Germain et al. (2015). These masks enforce the autoregressive dependency: the network computing parameters for $x_i$ has connections only from inputs $x_j$ where $j < i$.

The *generation* pass (from $\mathbf{z}$ to $\mathbf{x}$) is given by the reverse operation:

$$x_i = z_i \cdot \sigma_i(\mathbf{x}_{1:i-1}) + \mu_i(\mathbf{x}_{1:i-1}) \tag{5}$$

This inversion is sequential, and constructs each component of $\mathbf{x}$ one by one, requiring $\mathcal{O}(d)$ calls to the model, which can be slow for sampling. However, density evaluation (inverse pass) is parallelizable for training. The Jacobian $J_{T^{-1}}$ is triangular with diagonal entries determined by $\sigma_i$. Its determinant is simply $\prod_{i=1}^{d} 1/\sigma_i(\mathbf{x}_{1:i-1})$, which is computationally simple.

**Remark.** *The above description applies to one single layer of MAF transformations, and forces a single order of $i$ being dependent on earlier components, which is not suitable in general. In practice, one applies a permutation after each block to allow a component $i$ to depend on other components as well. A permutation is volume preserving and adds nothing to the Jacobian of the overall transform.*

### TRANSFORMER-BASED AUTOREGRESSIVE FLOW (TARFLOW)

TARFlow(Zhai et al., 2024) is a recent version of the autoregressive flow model, that intends to utilize the Transformer (Vaswani et al., 2017) architecture's superior capacity for modeling long-range dependencies and complex interactions among the components, increasing the expressiveness of the flow. TARFlow retains the fundamental MAF equations (Eqn 4,5). The main difference lies in the function approximator for $\mu_i$ and $\sigma_i$.

Instead of a masked feedforward network as in MADE (Germain et al., 2015), TARFlow (Zhai et al., 2024)uses a **causal Transformer encoder** to process the input vector $\mathbf{x}$.

1. **Input Representation:** The data vector $\mathbf{x}$ is embedded, often with positional encodings to inform the model of the variable ordering crucial for autoregressive structure.

2. **Causal Self-Attention:** The Transformer employs masked self-attention, where the query for position $i$ can only attend to keys and values from positions $j \le i$. This strictly enforces the autoregressive constraint, predictions for $x_i$ are based only on preceding elements, and automatically makes the Jacobian triangular.

3. **Parameter Prediction:** The output hidden state at position $i$ from the Transformer layer is fed into a small head network (e.g., a linear layer) to predict the parameters $\mu_i$ and $\sigma_i$ for that dimension.

The overall TARFlow model consists of stacking multiple such Transformers together (along with reversals) after each layer to allow information to flow among all components.

## 2.2 RELATED WORKS

**General Foundation Models**: The success of large-scale pretraining on language and vision tasks has inspired the development of foundation models for single-cell transcriptomics. Models such as Geneformer (Cui et al., 2024b), scBERT (Yang et al., 2022), scGPT (Cui et al., 2024a) are trained on vast atlases of cellular expression profiles. Their objective is to learn general-purpose representations of genes and cells that can be efficiently transferred to diverse tasks with minimal additional training. However, recent studies have highlighted their limitations in precisely modeling perturbational effects (Ahlmann-Eltze et al., 2025; Csendes et al., 2025).

**Perturbation Modeling**: Various different deep learning models have been developed to predict the outcomes of cellular perturbational intervention (Gavriilidis et al., 2024). These models often incorporate explicit inductive biases about biological systems. CPA (Lotfollahi et al., 2023) learns a joint latent space for genes and conditions. GEARS (Roohani et al., 2022) attempts to integrate gene-gene interaction networks directly into a graph-neural network architecture (Scarselli et al., 2008). PerturbNet (Yu et al., 2025) looks at probabilistic models for uncertainty quantification.

**Generative Models**: The advancements in diffusion models (Rombach et al., 2022; Podell et al., 2023) has also led to rise in similar approaches for perturbation modeling. Various models such as scDiffusion (Luo et al., 2024), CellFlow (Palma et al., 2024), and DiffDag (Shankar, 2025), have used diffusion models to learn transport from control cells to perturbed cell expressions. Recently, scDFM (Anonymous, 2026) have proposed a robust flow matching style generative model for perturbation prediction.

## 3 CELLTARNET

Consider $\mathcal{G}$ to be the set of genes under consideration; we will indexed these as $1..G$. A given cell can then be represented as a vector $x$ in $\mathbb{R}^G$, where component $i$ of the vector captures the expression level of gene $g_i$. A paired dataset consists of tuples $(x_{pre}, x_{post}, c)$ where $x_{pre}$ denotes the expression levels in the unperturbed (or control) cell, and $x_{post}$ represents the expressions in the post-perturbation cell. The perturbation used is encoded as a binary vector $c \in [0,1]^k$ where $k$ represents the set of perturbations. The general goal of in-silico perturbation modelling is to learn to predict $x_{post}$ based on the perturbation (ideally generalizing to unseen perturbations). We will focus on learning a generative model $p_\theta$ for the post-perturbation expressions based on the control cell and the perturbation i.e. $p(x_{post}|x_{pre}, c)$.

The proposed model is a conditional generative model designed to learn the complex distribution $p(\mathbf{x}_{post} \mid \mathbf{x}_{pre}, \mathbf{c})$. The architecture consists of two core components: a **Transformer-based Context Model** that encodes the conditioning context $(\mathbf{x}_{pre}, \mathbf{c})$, and a **Transformer-based Autoregressive Flow (TARFlow)** that transforms a noise sequence into a sample, with the conditioning function used for generation process attending via cross-attention to the context model's outputs.

The model operates in the following manner during sampling: For a given conditioning pair $(\mathbf{x}_{pre}, \mathbf{c})$, the conditioner produces a context representation. Simultaneously, a sequence of independent random variables $\mathbf{z} = [z_1, \ldots, z_T] \sim \mathcal{N}(0, I)$ is sampled from the base distribution. The flow model $f_\theta$ then performs an autoregressive, invertible transformation: $\mathbf{x}_{post} = f_\theta^{-1}(\mathbf{z}; \mathbf{x}_{pre}, \mathbf{c})$. This transformation is implemented as a sequence of transformer decoder blocks where cross-attention layers attend to the conditioner's output.

### 3.1 TRANSFORMER-BASED AUTOREGRESSIVE FLOW (TARFLOW)

The context model is a standard Transformer encoder $\mathcal{T}_\phi$. Its input is the concatenated and projected sequence:
$$\mathbf{S}_{cond} = [\mathbf{E}_{pre}\mathbf{x}_{pre} \; ; \; \mathbf{E}_c\mathbf{c}],$$
where $\mathbf{E}_{pre}$ and $\mathbf{E}_c$ are learned linear projections. A learnable [CLS] token is prepended to this sequence, along with learnable position embeddings. The encoder processes this input and outputs a sequence of contextualized embeddings $\mathbf{H}_{cond} \in \mathbb{R}^{(G+2) \times d}$. The hidden state of the [CLS] token serves as a global context summary, while the full sequence provides structured, position-aware features for cross-attention.

The generative model $f_\theta$ is implemented as a stack of $N$ modified Transformer blocks, with each block consisting of $K$ layers forming a neural autoregressive flow. The flow operates on a sequence $\mathbf{s}^{(0)} = \mathbf{z}$ (the noise input).

Each block $i$ first applies a permutation to its input. Each layer $l$ in the block then implements the following sequence of operations, producing $\mathbf{s}_i^{(l)}$ from $\mathbf{s}_i^{(l-1)}$:

1. **Multi-Head Cross-Attention (MCA):** Perform cross-attention from the flow sequence to the context representation after applying normalization:

$$\tilde{\mathbf{Z}}^{(l)} = \text{MCA}(LN(\mathbf{s}_i^{(l-1)}), \mathbf{S}_{cond}) + \mathbf{s}_i^{(l-1)}$$

   where the MCA operation is defined as:

$$\text{MCA}(\mathbf{Q}, \mathbf{KV}) = \text{Attention}(\mathbf{W}_Q\mathbf{Q}, \mathbf{W}_K\mathbf{KV}, \mathbf{W}_V\mathbf{KV})$$

   with $\mathbf{Q} = \mathbf{s}_{\text{norm}}^{(l-1)}$ (queries from the flow) and $\mathbf{KV} = \mathbf{S}_{cond}$ (keys and values from the context). For autoregressive flow modeling, we use *full attention* in this cross-attention step, allowing each position in $\mathbf{s}$ to attend to all positions in $\mathbf{S}_{cond}$.

2. **Causal Multi-Head Self-Attention (MSA):** Apply masked self-attention (after normalization) to capture dependencies within the flow sequence:

$$\mathbf{Y}^{(l)} = \text{MSA}(LN(\tilde{\mathbf{Z}}^{(l)})) + \tilde{\mathbf{Z}}^{(l)}$$

   where the MSA operation uses a causal mask $M$ to ensure autoregressive generation:

$$\text{MSA}(\mathbf{X}) = \text{Attention}(\mathbf{W}_Q\mathbf{X}, \mathbf{W}_K\mathbf{X}, \mathbf{W}_V\mathbf{X}, M)$$

   with $M_{i,j} = -\infty$ for $j > i$ (positions cannot attend to future positions).

3. **Multilayer Perceptron (MLP):** Apply position-wise feedforward transformation with GeLU based activation:

$$\mathbf{Z}^{(l)} = \text{MLP}(\mathbf{Y}_{\text{norm}}^{(l)}) + \mathbf{Y}^{(l)}$$

This ordering–cross-attention first, then self-attention–allows the conditioning information from $(\mathbf{x}_{\text{pre}}, \mathbf{c})$ to first modulate the flow representation before the self-attention captures dependencies within the generated sequence.

For autoregressive modeling, we use the output of each block to parameterize affine transformations. After each TarFlow layer $l$, we extract affine parameters from the transformed representation. Specifically, from $\mathbf{s}_i^{(K)}$, we compute a per position output:

$$[\boldsymbol{\mu}, \boldsymbol{\alpha}] = \text{MLP}_{\text{affine}}(\mathbf{Z}^{(l)})$$

where $\boldsymbol{\mu}, \boldsymbol{\alpha}$ are the shift and log-scale parameters. These define the affine parameter transformation for the input of the transformer block. i.e. flow transformation between layers is :

$$\mathbf{s}_{i+1}^{(0)} = \mathbf{s}_i^{(0)} \odot \exp(\boldsymbol{\alpha}) + \boldsymbol{\mu}$$

with the log Jacobian determinant contribution:

$$\mathcal{J} = \sum_{t=1}^{T} \sum_{j=1}^{d} \boldsymbol{\alpha}_{t,j}^{(l)}$$

Training maximizes the log-likelihood via the change-of-variable formula:

$$\mathcal{L}_{\text{NLL}} = -\mathbb{E}_{(\mathbf{x}_{\text{post}}, \mathbf{x}_{\text{pre}}, \mathbf{c}) \sim \mathcal{D}} \left[ \log p_Z(f_\theta(\mathbf{x}_{\text{post}}; \mathbf{x}_{\text{pre}}, \mathbf{c})) + \mathcal{J} \right].$$

### 3.1.1 DENOISING REGULARIZATION FOR CONDITIONING ROBUSTNESS

To prevent the flow from ignoring the $\mathbf{x}_{\text{pre}}$ input and relying solely on the noise and perturbation label $\mathbf{c}$, a specific a form of mode collapse, we introduce a dual denoising training task.

For each training tuple $(\mathbf{x}_{\text{post}}, \mathbf{x}_{\text{pre}}, \mathbf{c})$:

1. **Primary Task:** We create a noisy pre-state $\tilde{\mathbf{x}}_{\text{pre}} = \mathbf{x}_{\text{pre}} + \boldsymbol{\epsilon}, \boldsymbol{\epsilon} \sim \mathcal{N}(0, \Sigma_{diag})$ where $\Sigma_{diag}$ is the diagonal of the covariance of the expression in the training data. The model is trained to predict the true $\mathbf{x}_{\text{post}}$ from the *noisy* context $(\tilde{\mathbf{x}}_{\text{pre}}, \mathbf{c})$, yielding loss $\mathcal{L}_{\text{NLL}}^{\text{post}}$.

2. **Auxiliary Denoising Task:** Using the *same* model weights, we also train it to denoise the pre-state. Here, the input is the noise sequence $\mathbf{z}$, the target is the clean $\mathbf{x}_{\text{pre}}$, and the condition is $(\tilde{\mathbf{x}}_{\text{pre}}, \mathbf{c} = \mathbf{0})$ (i.e., a null perturbation). This yields loss $\mathcal{L}_{\text{NLL}}^{\text{pre}}$.

The combined flow loss is:
$$\mathcal{L}_{\text{Flow}} = \mathcal{L}_{\text{NLL}}^{\text{post}} + \lambda_{\text{denoise}} \cdot \mathcal{L}_{\text{NLL}}^{\text{pre}}.$$
This auxiliary task forces the conditioner's Transformer encoder to learn a robust, noise-invariant representation of $\mathbf{x}_{\text{pre}}$, which is then indispensable for the primary generation task.

### 3.1.2 CONTRASTIVE LEARNING FOR DISTRIBUTIONAL ROBUSTNESS

Maximizing likelihood alone can lead to a model that produces an overly compact distribution, underestimating skewness and spread or excessive mode covering behaviour. To promote learning of the distributional nuances, we add a contrastive objective in the latent space of the flow. We leverage the fact that the flow defines a bijection between a data point $\mathbf{x}_{\text{post}}$ and its latent code $\mathbf{z} = f_\theta(\mathbf{x}_{\text{post}}; \mathbf{x}_{\text{pre}}, \mathbf{c})$. A well-formed conditional distribution should map samples from the same conditional distribution to a coherent region in $\mathcal{Z}$-space, and separate them from samples from other distributions.

**Positive/Negative Pair Construction**

- **Anchor:** A latent code $\mathbf{z}_i = f_\theta(\mathbf{x}_{\text{post}}^{(i)}; \mathbf{x}_{\text{pre}}^{(i)}, \mathbf{c}^{(i)})$.

- **Positive:** The latent code from a *different* sample $\mathbf{x}_{\text{post}}^{(j)}$ that shares the **same pre-state and perturbation** $(\mathbf{x}_{\text{pre}}^{(i)}, \mathbf{c}^{(i)})$.

- **Negatives:** Two types:

   1. **Different Context, Same Post-State:** Latent codes for the *same* $\mathbf{x}_{\text{post}}^{(i)}$ but conditioned on a *different* context $(\mathbf{x}_{\text{pre}}^{(k)}, \mathbf{c}^{(k)})$ where $\mathbf{c}^{(k)} \neq \mathbf{c}^{(i)}$.

   2. **Same Context, Different Post-State:** Latent codes for a *different* post-state $\mathbf{x}_{\text{post}}^{(l)}$ drawn from the *same* batch, conditioned on the *anchor's context* $(\mathbf{x}_{\text{pre}}^{(i)}, \mathbf{c}^{(i)})$.

Let $g_\psi$ be a projection head that maps latent codes to an embedding space. The InfoNCE contrastive loss (Oord et al., 2018) for anchor $i$ is:
$$\mathcal{L}_{\text{Cont}}^{(i)} = -\log \frac{\exp(\text{sim}(\mathbf{e}_i, \mathbf{e}_i^+)/\tau)}{\exp(\text{sim}(\mathbf{e}_i, \mathbf{e}_i^+)/\tau) + \sum_{m \in \mathcal{N}_{\text{ctx}}(i)} \exp(\text{sim}(\mathbf{e}_i, \mathbf{e}_m)/\tau) + \sum_{n \in \mathcal{N}_{\text{post}}(i)} \exp(\text{sim}(\mathbf{e}_i, \mathbf{e}_n)/\tau)},$$
where $\mathbf{e} = g_\psi(\mathbf{z})$, sim is cosine similarity, $\tau$ is temperature, and $\mathcal{N}_{\text{ctx}}, \mathcal{N}_{\text{post}}$ are the sets of context and post-state negatives. The total loss is $\mathcal{L}_{\text{Cont}}$ is just the average over samples.

This objective encourages the flow's latent space to be semantically structured: points conditioned on the same context are pulled together, while impostors are pushed away, leading to a richer, more calibrated output distribution.

The final objective for training the model end-to-end is:
$$\mathcal{L}_{\text{Total}} = \mathcal{L}_{\text{Flow}} + \lambda_{\text{cont}} \cdot \mathcal{L}_{\text{Cont}},$$
where $\lambda_{\text{cont}}$ balances the contrastive regularization. This framework ensures the model learns a powerful conditional generative process that is faithful to the precise input context and capable of capturing the inherent stochasticity and multimodality of cellular perturbation responses.

## 3.2 Gene Networks as Attention Prior

Transformers are known to attend on weak features and correlations, especially when dealing with data much smaller than their expression capacity (Correia et al., 2019). This is exacerbated in high dimensional conditioning models, as one often has very few samples for a given condition variable (Roy et al., 2021). While contrastive learning can reduce this to an extent (Parulekar et al., 2023), a more natural way to fix this is to use knowledge priors (Roy et al., 2021). This is particularly important in perturbation modeling, where sparse regulatory networks often determine the gene expression (Aguirre et al., 2025).

Using transformer based conditioning in TARFlow models provides a natural way to incorporate such priors. Specifically given some form of dependence graph, we can use that along with the masking operation to prevent genes from attending on irrelevant factors. While in an ideal world, we might provide the exact gene dependence network, more realistically one needs to infer it from data (Young et al., 2016). Following existing works, we use the correlation matrix of gene expressions. Specifically given the correlation matrix, we use the top-k highest correlated (positive or negative) genes for each gene. Next we additionally use a low-rank approximation of the correlation matrix to determine a second subset of genes of genes to attend for each gene. Earlier work has shown that such L+S methods are better than individual methods (Chang et al., 2015). We use a binarized L+S matrix to construct the sparse attention matrix used in the Self-Attention parts of the transformer in the flow model (not the context model).

## 4 Experiments

**Evaluation Protocol**   We employ the Norman combinatorial gene-perturbation dataset (Norman et al., 2019), which comprises CRISPR-based overexpression perturbations in the K562 cell line, including both single-gene and paired double-gene perturbations. To rigorously test generalizability, we adopt the "harder hold-out" protocol, wherein specific paired perturbation combinations are systematically excluded from the training data (Lopez et al., 2023). The model's performance is then assessed on its ability to predict outcomes for these unseen combinatorial perturbations, simulating a realistic scenario where not all possible combinations can be empirically profiled.

As no standard train-test split is established for this benchmarks, we perform an evaluation using 10-fold cross-validation, reporting aggregated performance across all folds. For preprocessing the data we follow the protocol described by Ahlmann-Eltze et al. (2025).

**Baselines**   We benchmark our method against a suite of baselines models, encompassing several different approaches. These include: CPA (Lotfollahi et al., 2023) an autoencoder-based model integrating perturbation and covariate information; GEARs (Roohani et al., 2022) a graph-neural network approach explicitly modeling gene interactions; the foundation models Geneformer (Cui et al., 2024b), scGPT (Cui et al., 2024a) and the more recent approaches of Cellflow (Palma et al., 2024) and STATE (Adduri et al., 2025).

**Evaluation Metrics**   Performance is quantified across three complementary axes to provide a holistic view of model capability:

- Pointwise Reconstruction Accuracy: Measured via standard regression metrics: Mean Absolute Error (MAE), Mean Squared Error (MSE), and the Pearson correlation coefficient between predicted and observed expression values for each gene. Following Viñas Torné et al. (2025), we also report Pearson $\hat{\Delta}$, which adjusts for the control–perturbation baseline bias.
- Perturbation Specificity: Evaluated using the perturbation discrimination score (PDS) (Wu et al., 2024; Liu et al., 2025). This metric assesses whether the model correctly produces distinct expression profiles for distinct perturbation inputs, emphasizing specificity.
- Effect Ranking: Assessed via the Spearman rank correlation between the predicted and true gene expression profiles. This metric evaluates the model's ability to correctly capture the relative ordering of gene expression levels, which is important for downstream biological interpretation.

**Results on Norman Dataset**    Our model demonstrates strong overall performance, achieving the best scores across most metrics (reported in Table 1). It records the lowest reconstruction metrics (L2, MSE, MAE) outperforming the next-best baselines by approximately 4.5%, 4%, and 3%, respectively. Our model also does better on the PDS score as well as the adjusted Pearson $\Delta$. We additionally see that GEARS does well on correlation metrics but does not do good reconstruction. scGPT is both bad at reconstruction and prediction direction correlation (DE-Spearman). The recent STATE model is quite poor at reconstruction. Finally, Geneformer generally seems to be a competitive baseline when it comes to generalization.

Table 1: Comparison of different methods across evaluation metrics on Norman holdout. Control refers to the mean expression from control cells

| Model | L2 $\downarrow$ | MSE $\downarrow$ | MAE $\downarrow$ | DE-Spearman $\rho$ $\uparrow$ | Pearson $\Delta$ $\uparrow$ | PDS $\uparrow$ | Pearson $\hat{\Delta}$ $\uparrow$ |
|---|---|---|---|---|---|---|---|
| Control | 4.9272 | 0.02495 | 0.05149 | - | - | 0.6471 | 0.1552 |
| scGPT | 4.3909 | 0.01911 | 0.04324 | -0.0801 | 0.6620 | 0.6797 | 0.3369 |
| GEARS | 4.3540 | 0.01886 | 0.08456 | 0.4484 | 0.8719 | 0.8046 | 0.3007 |
| Geneformer | 2.4677 | 0.00612 | 0.02862 | 0.5049 | 0.8857 | 0.8879 | 0.7353 |
| CPA | 7.0673 | 0.04314 | 0.09674 | 0.4212 | 0.4804 | 0.7303 | 0.2924 |
| STATE | 22.8098 | 0.3958 | 0.3304 | 0.4903 | 0.0076 | 0.6598 | -0.0026 |
| CellFlow | 2.5052 | 0.00547 | 0.02852 | 0.6098 | 0.8361 | 0.8937 | 0.7809 |
| Ours | **2.3581** | **0.00526** | **0.02765** | **0.7089** | **0.9103** | **0.9066** | **0.8356** |

## 5    CONCLUSION

We present a model for conditional modeling of gene expressions response to perturbations based on paired single cell dataset. The two major insight of our model are: a) to utilize contrastive training as additional supervision during generative modeling; and b) to use a matrix based on co-occurrence to regularize attention during sample generation. The former allows our model to successfully captures the full distributional shift induced by a perturbation, while also learning to seperate the effects of different perturbations. The latter allows the model's attention mechanism to be guided by plausible gene-gene interactions, improving generalization to unseen perturbations. Initial experiments on genetic perturbation experiments show strong prediction performance, especially on generalizing to unseen predictions.

## MEANINGFULNESS STATEMENT

This work helps us learn meaningful representations of life by modeling how cells, the fundamental units of life, respond to change. Our method predicts the full distribution of a cell population's response to a perturbation, allowing it capture the biological complexity of heterogeneous cellular response to treatments. By incorporating approximation of biological networks into the model, we ensure the learned representations reflect genuine gene regulatory mechanisms.

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
