# OpenReview forum: "CELLTARNET: SINGLE-CELL PERTURBATION PREDICTION USING TRANSFORMER BASED NORMALIZING FLOW"
_ICLR.cc/2026/Workshop/LMRL — ICLR 2026 Workshop LMRL Poster_

### Official Review · Reviewer_pjtk · 2026-02-22
**technically solid, with strong empirical results**

**Rating:** 7
**Confidence:** 3

**Review:**

This paper proposes CellTARNet, a conditional Transformer-based normalizing flow model for predicting post-perturbation single-cell gene expression distributions. The approach combines a Transformer autoregressive flow backbone (TARFlow), with a contrastive objective to structure conditional distributions, and a biologically motivated sparse attention derived from gene-gene correlation structure.

Strengths:
- Well-motivated focus on modeling full distributions rather than mean shifts.
- Thoughtful integration of contrastive learning with normalizing flows.
- Inductive bias via gene-network–guided attention is biologically sensible.
- Strong performance on one combinatorial perturbation dataset, outperforming several competitive baselines across reconstruction and ranking metrics.

Weaknesses:
- Evaluation is limited to a single dataset (Normal et al)
- Lack of ablation studies isolating the contributions of contrastive loss and attention priors.
- Despite emphasizing distributional modeling, evaluation lacks distributional metrics such as Wasserstein
- Results are basically one Table, lack of convincing Figures showing for example predicted vs. ground truth distribitions of cells in UMAP space or any other useful visualization
- Also, the paper would greatly benefit from a Figure describing the architecture.
- Some clarity and reproducibility details are missing.

Overall: A technically solid and relevant workshop contribution with strong empirical results, though additional ablations and broader validation would strengthen the claims.

---

### Official Review · Reviewer_gntk · 2026-02-23
**TARNet for perturbation response prediction**

**Rating:** 6
**Confidence:** 3

**Review:**

Authors suggest CellTARNet, a TARNet-based single-cell perturbation response prediction method with specific modifications, namely by adding contrastive loss and denoising loss, and using the gene expression correlation as masks of attention layers of TARNet’s transformers. The authors apply the method to the Norman 2019 dataset to show the superior performance metrics over other perturbation response prediction methods. While the stated performance seems good, too many details on the experiments are missing to gauge if this result is prudent.

- Ablation studies are missing, thus it is unclear which part of the method’s modifications contributed to the performance.

- Major details are missing.
  - How are conditions encoded? If the CLS token is learned, what is being used for combinatorial perturbation where neither of the pairs was observed by the model?
  - How is the order of genes selected? Would the performance differ when the order is shuffled? Were predicted genes selected as the differentially experessed ones? How many genes were used, and how long did it take to train on such a number of genes?
  - What were the settings the benchmark methods were run with? How are their and CellTARNet's hyperparameters selected?
  - From which dataset was the correlation matrix of gene expression for the attention network obtained? It should have been from unperturbed K562 or independent datasets, rather than perturbed data, as in this case, the model has already partially seen the outcome.

- Contrastive objective 1 (different context, same post-state) is not necessary. The inverse maps just need to be different in a distributional sense, and I wonder if this constraint is harming the performance- e.g., if a cell is in a state that is not responsive to any of the perturbations, they need to be artificially separated into different groups in pre-state space with this contrastive loss.

- As MAE highly depends on the kernels it uses, the authors should measure Sinkhorn divergence. All other metrics are the mean statistics.

- Perhaps a minor comment: when reconstructing $x_{pre}$, why was only the independent noise was added? I.e., why not use the noise with the same covariance?

- I would be interested to see a discussion about why the autoregressive flow approach, which is an unnatural constraint on unordered gene expression, works better than other methods that do not have that constraint.

Overall, it is difficult to fully understand how the experiments were conducted, thus difficult to gauge the credibility of the experimental result from what is stated in the manuscript. However, if significant details and benchmarks across hyperparameters and ablations can be added, this work suggests an alternative generative model for single-cell perturbation prediction. It would be interesting to compare this approach with CellFlow and STATE in terms of compute time, training stability, and predictive fit given differing generative modeling strategies and constraints.

---

### Official Review · Reviewer_FAp5 · 2026-02-24
**Sound method, inconsistent evaluation and presentation.**

**Rating:** 4
**Confidence:** 5

**Review:**

**1. Summary**

This paper proposes CellTarNet, a conditional generative model for predicting single-cell transcriptional responses to genetic perturbations. The architecture combines a transformer-based normalizing flow (TARFlow) with a transformer encoder that conditions generation on pre-perturbation expression and perturbation identity. The authors introduce three auxiliary components: a denoising regularization task to prevent the model from ignoring pre-perturbation context, an InfoNCE contrastive loss in the flow's latent space to encourage distributional calibration, and a sparse attention mask derived from gene expression correlation matrices to inject biological priors. The model is evaluated on the Norman et al. (2019) combinatorial perturbation dataset using a held-out split protocol and compared against several baselines including CPA, GEARS, scGPT, Geneformer, CellFlow, and STATE.

**2. Strengths**

The use of cross-attention from the flow to a context encoder is an architecturally clean design for conditional generation, and it provides a natural mechanism for incorporating perturbation identity alongside pre-perturbation state. The denoising regularization (Section 3.1.1) is a practical contribution. It addresses a genuine failure mode: the risk that the generative model ignores $x_\text{pre}$ and collapses to a perturbation-label-conditioned prior. The evaluation protocol adopts the harder held-out split for combinatorial perturbations, which tests a meaningful notion of generalization. The reported results in Table 1 show consistent improvements over baselines across most metrics. Related to the former, the concurrent usage of all the models presented in the evaluation is an engineering feat in and of its own.

**3. Weaknesses**

* **3.1** Disconnect Between Distributional Claims and Point-Wise Evaluation The paper's central motivation emphasizes modeling "the full distribution of perturbation effects," including heterogeneity, bifurcating fates, and multimodality (Abstract, Section 1). However, every evaluation metric in Table 1 is a point-wise reconstruction measure (MSE, MAE, Pearson correlation, Spearman correlation). No distributional metrics are reported e.g. Wasserstein distance, maximum mean discrepancy or similar. The Perturbation Discrimination Score (PDS) measures specificity across perturbations, not distributional fidelity within a perturbation. It’s difficult to support the paper’s primary claims without distributional evaluation.
* **3.2** No Ablation Study The model introduces at least three distinct components on top of the base TARFlow: contrastive loss, denoising regularization, and correlation-based attention masking. A single, final score is provided; no ablation is in place to determine which components drive performance. It is therefore unclear whether the observed gains are attributable to the architectural innovations claimed, increased model capacity or similar “horizontal” modeling choices.
* **3.3** The Contrastive Objective Has a Formulation Gap Positive pairs are defined as latent codes from "a different sample $x_\text{post}$ that shares the same pre-state and perturbation" (Section 3.1.2). However, in single-cell perturbation experiments the same cell is never observed before and after perturbation. Two cells cannot literally share the same $x_\text{pre}$. The authors likely intend cells from the same perturbation condition; I may be mistaken. However, as it stands, this is not what the formulation states. This ambiguity undermines confidence in the contrastive setting: is the positive/negative construction truly achievable with real data?
* **3.4** Gene Correlation as "Biological Knowledge" is Weakly Justified The attention prior (Section 3.2) is derived from the expression correlation matrix of the training data, described as incorporating "structured biological knowledge." Expression correlation is not equivalent to regulatory interaction; it conflates direct regulation, indirect co-regulation, and technical confounds (in the form of batch effects or similar). No analysis is provided of what the resulting attention masks capture, whether they align with known regulatory relationships, or whether they improve performance relative to full attention (which is again an ablation concern).
* **3.5** Results Lack Variance Estimates and Statistical Context The paper reports 10-fold cross-validation but presents only point estimates in Table 1. No or significance tests, confidence intervals or standard deviations. Without variance estimates, it is impossible to assess whether the reported improvements (e.g., ~4% in L2 over Geneformer) are statistically meaningful or within noise. Perturbation prediction benchmarks typically exhibit high variance across folds, making this analysis crucial, and its absence that much more apparent.
* **3.6** Excessive Preliminary Material at the Expense of Analysis Section 2.1 devotes approximately 1.5 pages to textbook exposition of normalizing flows and masked autoregressive flows. While the summaries provided are good, the material is standard and does not require re-derivation for the LMRL audience. This could easily go into the appendix. I’d argue the space would be better used for the aforementioned ablation experiments, distributional evaluation, or analysis of the attention priors.
* **3.7** Citation and Notation Issues The Geneformer citation (Cui et al., 2024b) appears to reference a gene compression paper rather than the Theodoris et al. (2023) foundation model. Notation is inconsistent: G denotes both the gene set and its cardinality; c is introduced as a binary vector but subsequently treated as a scalar label.  In general, manuscript presentation could be improved. A figure noting the proposed architecture would be nice to see!

**4. Relation to Prior Work**

The paper positions itself against foundation models (scGPT, Geneformer) and perturbation-specific architectures (CPA, GEARS, CellFlow). The comparison is reasonable in scope, though the absence of the recent scDFM model (which the authors cite as concurrent work) from the benchmark is notable. The relationship to CellFlow deserves deeper discussion. In particular, I’d appreciate a dive on what architectural differences account for the performance gap reported in Table 1.

**5. Questions for the Authors**

* Can the authors provide ablation results isolating the contributions of the contrastive loss, denoising regularization, and attention masking?
* How does the model perform under distributional metrics (e.g., MMD, Wasserstein distance) that directly assess the fidelity of the predicted response distribution?
* How are positive pairs in the contrastive objective constructed in practice, given that individual cells do not have matched pre/post observations? Is there a common condition baseline?
* What is the computational cost relative to CellFlow and GEARS, and how does sampling scale with gene dimension given the sequential nature of autoregressive generation?

**6. Overall Assessment**

The paper addresses a relevant problem and proposes an architecture that combines several reasonable ideas: conditional normalizing flows, contrastive regularization, and attention biasing via gene networks. However, the current submission has substantial gaps between its claims and their substantiation. The distributional modeling motivation is not supported by distributional evaluation; the individual contributions are not isolated via ablation; the contrastive formulation contains an apparent inconsistency with the data-generating process; and the results lack variance estimates. In its current form, the methodological contributions remain plausible but unvalidated, and the paper would benefit substantially from ablation experiments, distributional metrics, and greater care in technical exposition before it can support its claims.

---

### Meta-Review · Area_Chair_6Zoa · 2026-02-25

**Recommendation:** Accept (Poster)
**Confidence:** 3

**Metareview:**

Accept/

---

### Decision · Program_Chairs · 2026-03-02

**Decision:**

Accept (Poster)

**Comment:**

Please see the meta-review.